# Harvesting the Aggregate Computing Power of Commodity Computers for Supercomputing Applications

**Dereje Regassa** [1] , **Heonyoung Yeom** [1] **and Yongseok Son** [2,*]

1 Department of Computer Science and Engineering, Seoul National University, Seoul 08826, Korea; dereje@snu.ac.kr (D.R.); yeom@snu.ac.kr (H.Y.)
2 Department of Computer Science and Engineering, Chung-Ang University, Seoul 06974, Korea
* Correspondence: sysganda@cau.ac.kr

**Abstract:** Distributed supercomputing is becoming common in different companies and academia. Most of the parallel computing researchers focused on harnessing the power of commodity processors and even internet computers to aggregate their computation powers to solve computationally complex problems. Using flexible commodity cluster computers for supercomputing workloads over a dedicated supercomputer and expensive high-performance computing (HPC) infrastructure is cost-effective. Its scalable nature can make it better employed to the available organizational resources, which can benefit researchers who aim to conduct numerous repetitive calculations on small to large volumes of data to obtain valid results in a reasonable time. In this paper, we design and implement an HPC-based supercomputing facility from commodity computers at an organizational level to provide two separate implementations for cluster-based supercomputing using Hadoop and Spark-based HPC clusters, primarily for data-intensive jobs and Torque-based clusters for Multiple Instruction Multiple Data (MIMD) workloads. The performance of these clusters is measured through extensive experimentation. With the implementation of the message passing interface, the performance of the Spark and Torque clusters is increased by 16.6% for repetitive applications and by 73.68% for computation-intensive applications with a speedup of 1.79 and 2.47 respectively on the HPDA cluster. We conclude that the specific application or job could be chosen to run based on the computation parameters on the implemented clusters.

**Keywords:** HPC; shared memory; optimization; commodity hardware; big data





## 1. Introduction

High-performance computing (HPC) is defined in terms of distributed, parallel computing infrastructure with high-speed interconnecting networks and high-speed network interfaces, including switches and routers specially designed to provide an aggregate performance of many-core and multicore systems, computing clusters, in a cloud or the form of a grid. Whereas the grid spans a large area and uses WAN protocols, clusters are suitable for performing scientific computations in a small area. Cluster-based supercomputing can be classified into different types based on the hardware, inter-networking standards, devices (e.g., switches), number of sockets, number of cores in a rack, and so on. Recently, applications of HPC in scientific research have greatly increased as an alternative to dedicated supercomputers [1–3], because they are costly, difficult to program, and require sophisticated expertise. Supercomputers are generally stand-alone, and scalability is a problem in supercomputer-based data centers [4]. The HPC cluster-based supercomputing platforms have been successfully implemented in various universities [1,5]. A list of open-source cluster management software [6] and some of the commercially available schedulers are available in various domains.

The university environment, where the prime goal is the transition from isolated, stand-alone computers to an HPC-based cluster computing environment, is substantially

different from individual and production-constrained computational processes in many aspects. The study team expects that the transition from stand-alone laboratory experiments to computationally intensive industrial-grade experiments must be performed so that researchers can understand the establishment of the cluster environment from scratch, the node configuration, and the data network, rather than any specific hardware requirements. These seamlessly provide access to the working cluster using commodity computers connected by a high-speed network [1].

The expectations and utility of the cluster environment vary across cluster users. For example, a cluster user sometimes does not even need to know that the job is completed with the help of a backbone a supercomputing facility. Some users may need to know only the details of the available memory and the number of necessary cores. In contrast, expert users may need to know almost everything, including protocols, scheduling policies, privileges, priorities, and everything directly or indirectly relevant to optimizing application performance.

The approach of using commodity computers to form a cheap cluster is found in different universities that aim to upgrade their stand-alone lab sessions into powerful, scalable, and large-scale experiments, such as computational climate models, computational fluid dynamics, numerical weather prediction models, and many more.

In this study, we focus on obtaining expertise in commodity clusters prior to any purchases through surveying a primitive set of applications intended for demonstration. Once most of the users understand the platform, the underlying hardware can be upgraded to the actual HPC standards, which benefits data rates and computation power.

Unlike in industry, university research clusters are flexible and less demanding in terms of failure after effects. In the absence of a real-world cost of a crash or failure of an application running on the commodity cluster, laboratory experiments can tolerate the breakdown of experiments due to a node or power failure, exhaustive use of available memory, system shutdown without notice, and other issues. We identified different challenges while introducing cluster computing in a university compared to a production-grade industry setup. These challenges influence many design decisions, not only the cluster implementation but also the culture of students and instructors. The users move from a single-threaded programming paradigm to a large-scale, threaded, distributed, and parallel form of programs, a fundamental element of success in the cluster-based supercomputing facility. Training and workshops may need to be organized to introduce the available resources to the staff and students at the university to provide basic libraries to support their experiments. Hence, we develop a cost-effective distributed supercomputing platform from cheap computers around us which can be used by various researchers for different applications.

The contribution of this work is as follows:

- We analyze the existing HPC infrastructure design options, review existing cluster solutions, and identify significant challenges related to the hardware, software, and networks in the implementation of cluster computing.
- We review the existing cluster solutions implemented at different university level and advanced research laboratories.
- We implement a cluster from 30 commodity computers to aggregate Gigaflops of computing power and use an efficient scheduler to run jobs on the cluster infrastructure.
- perform scientific experiments on a newly implemented cluster with distributed optimization and machine learning for a showcase.
- compare the practical performance of the proposed system under different configurations, considering various algorithms and loads.

The rest of this paper is organized as follows. Section 2 describes the background and motivation. Section 3 presents the design and implementation of the proposed idea using a cluster of commodity computers. Section 4 shows the experimental evaluation and its results. Section 5 discusses the related work and Section 6 concludes the paper with recommendations for the use of the output on a similar scale.

## 2. Background and Motivation

Since its inception in 1991, HPCC provided vital enabling technologies and disseminated progress in the field to extend the lead in the field of HPC. Later, more agencies joined in a coordinated effort to develop HPC. Most development and progress work in HPC was made available on the HPCC website as annual reports. This initiative led to significant improvements in HPC research.

The HPC cluster is used in implementations and experiments to compute the simulated Gaussian density function to solve a density estimation problem based on shared memory and the parallel execution of the kernel density estimation algorithm on a graphic processing unit (GPU)-enabled system [4,7], the OpenMP library [8]. The study results reveal significant improvement in the time taken by the HPC cluster in estimating the density function. Although GPU-based computing provides easy access to higher teraflops with fewer machines, this setup is only suitable for parallel jobs, whereas fine-grain parallelism, synchronization of threads, and programming GPU-enabled systems are complex tasks. Therefore, several studies have been performed on the Intel Xeon processor-based HPC cluster. These clusters have the advantages of being easy to program, supporting SIMD and MIMD models, and providing low-cost alternatives to a few teraflops with the ease of scalability.

The floating-point operations (FLOPs) [9–11] are used in scientific computing community to measure the processing power of individual computers, different types of HPC, Grid, and supercomputing facilities. China has made significant progress in developing HPC systems in recent years. Since the development of the supercomputer by their national defense university, the nation's first win of the Gordon Bell Prize at Supercomputing 2016 (SC16) also represents an accomplishment in HPC applications. As a result, various academic institutions, private companies, and research groups have collaborate and contributed to the development of HPC on their premises [3,10,12–14]. Major applications of HPC are in data storage and analysis, data mining, simulation and modeling, scientific calculations, bioinformatics, big data challenges, and complex visualizations.

However, the purchase of a supercomputer for academic and collaborative research may cost an investment of millions of dollars and therefore most universities avoid directly purchasing the supercomputers. But to provide a platform for the students and researchers to work on the most challenging scientific problems related to different fields such as computational fluid dynamics, medical imaging, graphics, higher dimensional visualization, big data, parallel machine learning, and data mining multidisciplinary optimization. Many universities have implemented low-cost supercomputing using clusters of Intel processors. Commodity computers fail to process complex jobs with very large data and computations as memory requirements and millions of times complex function evaluations. To solve bigger problems, scientists have invented cost-effective solutions to aggregate individual gigaflops of individual computers by clustering and facilitating the high-speed I/O and communication to provide a platform that performs the large-scale experiments. In this study, most reviewed research ideas are taken from HPC applications in scientific computing because the objective is to provide an HPC-based supercomputing infrastructure for various scientific computing applications.

## 3. Design and Implementation

This study started with specific design goals and objectives restricted to the installation of a pilot HPC cluster for future research projects. This work brings the ideas of alternative options to develop a supercomputing environment from available commodity computers for those researchers who need a supercomputing platform for their research yet can not buy the HPC. We followed the standard methodology for information technology/information and communication technology projects during the implementation of DeepStack and high-performance-data analytics (HPDA) clusters. This study used a hybrid methodology involving qualitative, quantitative, and empirical research aspects at various stages. The two stages of this study are identified as (i) the design and imple-

mentation of the HPC clusters as a platform and (ii) the empirical performance measure of various parameters of the newly implemented platform. The first objective relies on qualitative research aspects aimed at the platform design, whereas the second stage focuses on quantitative and empirical performance measures under the careful design of relevant experiments.

As the industry estimates that we are creating Quintilian bytes of data every year which makes data processing, transmission, and storage difficult it has created entirely new sets of challenges and has forced us to find new ways to handle Big data effectively [15]. From document analysis of the existing HPC infrastructure [16–19], we choose the HPC environments that are expected to perform CPU-bound operations with a heavy asymmetric load on compute nodes. Therefore, we choose the Deep Stack cluster and HDPA-based HPC clusters as the computing power of the nodes as well as message passing infrastructure between the processes plays an important role in the performance of application programs.

In our design-oriented approach to the installation of the HPC cluster platform, the initially necessary data collection from various sources and experts has been done, including standards, benchmarks, heuristics, networks, topology, workload patterns, optimization methods, node configuration profiles, and detailed description of protocol selection procedures. The analyzed data were used for planning, topology design, network installation, and the actual design and implementation phase for the HPC cluster platform. After the platform was successfully set up, the empirical methodology was used, and the design of experiments, performance analysis, and data analysis tasks was completed.

The following methodologies are followed during the design and implementation of this research:

- **Data Collection**—different data sources were used to collect necessary information regarding hardware, software details and design issues, and availability of cluster management systems. Data was collected from various websites related to university-level HPC cluster systems from Stanford University, FSU, MIT, and Yale. Expert interviews were performed at different levels of professionals in the fields of HPC to get specific requirements for the design and implementation of the proposed platforms.
- **Data pre-processing**—Once the data were collected and encoded in a standard format, the pre-processing of data was performed including checks for consistency, validity, missing values, errors, and updates.
- **Planning and design of HPC platforms**—Based on the input of various experts and available HPC installations in the industry, these design parameters for the proposed systems were set. This included selection of appropriate network topology, protocols, hardware, and software components to enable the HPC server and compute nodes each.
- **Implementation**—The actual implementation of two separate types of HPC clusters was initiated within separate initiatives of data science and intelligent systems research groups. The implementation procedure was divided into various phases as per the life cycle of an ICT/IT project. On each HPC server and compute-node internetworking, protocol configuration, installation, and configuration of the servers and client software were completed sequentially.
- **Verification and Validation**—the recent installation against the requirements and objectives of the proposed research are verified. The correctness of installation and availability of HPC cluster and system-level services of the new HPC system was checked.
- **Design of experiments**—as the platform was implemented on available hardware, we have designed structured workloads for various experiments and tested the performance of the underlying platform under different settings. Since it is experimental research, we have designed two groups of experiments—the control group and the experiment groups. In the control group experiment, we have used a standard algorithms like the FIFO scheduling algorithm with default parameters for the installed platform that will be used as a base for comparison, in the experiment group we have an experimental setup with varying platform parameters like the number of

cores, memory, virtual memory, default wall-times, network topology and allocated bandwidth. These experiments were performed on introductory level workloads and the optimum performance profile of the underlying platform was deducted.

- **Data Analysis**—after the execution of planned stages and experiments on real time data, data analysis was performed to understand the performance of the system, and interpretation and implications of various parameters changes in newly installed HPC clusters.

### 3.1. DeepStack Cluster

To save the readers from confusion regarding general torque specifications and our specific implementation we have given a unique name for our implementation of torque cluster as the DeepStack HPC cluster. This cluster includes three 5-node of identical Intel core i-7 clusters with approximately 15 TB storage and 120 GB of RAM in the experimental phase with specifications given in Table 1. However, to get a better result, we have projections of 30-core permanently connected compute nodes under DeepStack and 100 Ad-hoc compute nodes during runtime but not guaranteed. It is possible to extend the capabilities of DeepStack as a source of the highest computing power and a full-fledged implementation of the scientific libraries for various scientific jobs. Using Torque with this setup enabled us in achieving our objectives of running Torque with MPICH2 and OpenMPI for large-scale, distributed memory jobs over available processors from configured compute nodes, and support various scientific libraries.

**Table 1.** DeepStack Cluster Specifications.

| Parameters | Used Values |
|---|---|
| Master Machine | CPU 2.25 Ghz |
| | RAM—8 GB |
| | HDD 1 TB |
| | OS—CentOS 7 |
| Computing Nodes | CPU 2.25 Ghz |
| | RAM—8 GB |
| | HDD 1 TB |
| | OS—CentOS 7 |
| Libraries | netcdf/intel/3.6.3 |
| | netcdf/intel/4.1.1 |
| | Stata/11, R/intel/2.9.2, |
| | matlab/2017/b |
| Software Packages | Cent OS Enterprise Server 7, |
| | Torque 6.1.1,OpenMPI |
| | MPICH |
| Language Support | C++, Java, Fortran |

In this study, each Torque subcluster is implemented from scratch and involves installation and configuring various server-side dependencies and libraries. The DeepStack cluster architecture is given in Figure 1.

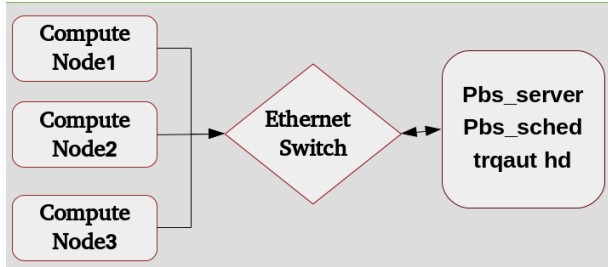

**Figure 1.** DeepStack cluster Architecture.

### 3.1.1. Resource Manager Components

Each torque resource manager designated as a server computer contains three components:

1.  Torque Server module (PBS_server)—In this architecture, each Torque server is given the same hostname and server name per the recommendation of Adaptive Computing Pvt. Ltd., Naples, FL USA, which developed Torque as open-source cluster management software. Each Torque server is an instance of PBS Server. Each Torque server comes with a default queue called the batch. Certain configuration steps should be done in sequence to ensure the server installs with its dependencies.
2.  Scheduler module (PBS_sched)—Torque provides a very basic built-in scheduler capable of scheduling jobs in first-in-first-out order in job-exclusive and shared mode on compute nodes. If this cluster is used for scientific simulations, it must be configured with advanced schedulers, such as MOAB and MUAI, which are not free like the default scheduler. The default scheduler is enough for the start cluster computing. Each server instance is configured with the default scheduler.
3.  Authentication server module (TRQAUTHD)—as described as trquat hd in Figure 1 —Torque provides an authentication server(TRQAUTHD) whose job is to allow only pre-configured clients to submit jobs. An instance of TRQAUTHD is required for the PBS_server along with the PBS_sched demon. By default, the server computer is allowed to submit jobs on the server, but the purpose of TRQAUTHD is to register the clients with a server so that jobs can be submitted by the external IP address registered using TRQAUTHD.

### 3.1.2. Torque Compute Node Components

The Torque compute node contains the PBS Multi operation machine (MoM) modules that participate in job execution in the processor exclusive and processor shared modes. If a compute node is enabled to submit jobs for execution, the TRQAUTHD client-side module must be configured with compute node.

1.  The PBS MoM module communicates with the PBS server and participates in the actual execution of jobs. It provides various resources for job execution, such as processors, memory, and wall-time (the actual time that a clock on the wall or a stopwatch in hand measures from the start of a process to the end of the process). Each MoM receives its jobs in a queue and executes one job at a time in basic settings. Each job acquires a lock over the MoM node and proceeds to completion before the processor is freed for other jobs on the server in the waiting queues. During the implantation, various communication patterns were noticed between the MoM node and PBS server, which will be presented as case studies to help researchers troubleshoot various issues in job executions, debugging, and performance enhancement.
2.  Authentication Module (TRQAUTHD)—each MoM node is allowed to submit jobs at a server and must have an authentication module (TRQAUTHD) installed and configured. For MoM nodes, it is an optional package.

### 3.1.3. Torque Client Node

The Torque client modules were developed at the time of installation of the Torque server and can be copied to clients with the necessary scripts for their registry in the system services. Torque clients are designated nodes that can submit jobs to the PBS server. Each client can be registered with the server by its hostname and IP address. A change in the IP and hostname of a client crash the job submission system, and the whole processes of registering and service invocation must be repeated to bring the system to the operating mode.

Each set of sub-clusters contains one server, an instance of the scheduler, and one authentication daemon at the server. For failover capability, identical servers are installed in similar clusters. Currently, five Compute nodes are attached to every server and one

client is also configured to submit the jobs. The complete architecture of the implemented cluster is given in Figure 1.

### 3.2. High-Performance Data Analytics Cluster

The low-cost, distributed, and data-intensive cluster, known as the HPDA cluster, has been set up in the designated laboratory. Distributed computing is just a distributed system where multiple machines perform specific work simultaneously. While performing the work, machines communicate with each other by passing messages between them. Distributed computing is practical when fast processing (computation) is required on big data. Apache Hadoop [20] and Apache Spark [21] are well-known examples of big data processing systems. The HPDA cluster comprises Apache Hadoop and Apache Spark. The cluster specifications and diagram are illustrated in Table 2 and Figure 2 respectively.

**Table 2.** HPDA Cluster Specifications.

| Parameters | Used Values |
|---|---|
| Number of Nodes:<br>Cores in Use:<br>Availed Memory:<br>Availed HDFS: | 10 (1 Master, 9 Slaves)<br>72<br>24.9 GB<br>6.4 TB |
| Master Machine | CPU 2.25 Ghz<br>RAM—4 GB<br>HDD 1 TB<br>OS—Ubuntu 16.04 LTS |
| Ethernet Switch | 16 Port 10/100 Mbp |
| Software Tools | Hadoop 2.7, Spark 2.1.0,<br>Scala 2.12.1, OpenJDK 1.8<br>Hive 2.0.0, HBase 1.1.4<br>Flume 1.6.0, Anaconda3 4.3.0<br>R 3.3.2, RStudio 1.0.136 |
| Language Support | Scala, Java, Python, R, &<br>MapReduce |

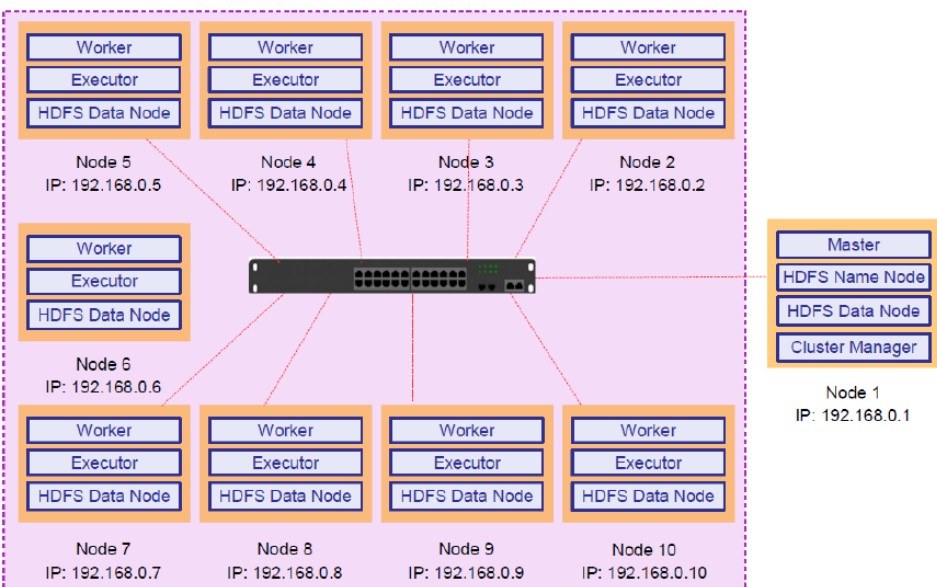

**Figure 2.** HPDA cluster diagram.

### 3.2.1. Apache Hadoop Setup

The Apache Hadoop software is an open-source framework built for reliable, scalable distributed computing tasks with huge data sets over a cluster of multiple computers. The features used for this implementation are (i) the large data set distribution across clusters of computers using a simple programming model (ii) became the de facto standard for storing, processing, and analyzing hundreds of terabytes and petabytes of data and (iii) is cheaper to use in comparison to other traditional proprietary technologies and can handle all type of data.

Generally, a Hadoop system comprises a computer acting as the master node and multiple computers acting as slave nodes, as shown in Figure 2. Hadoop has two modules, in total, including the HDFS and MapReduce Framework. The HDFS usually only has one NameNode, which manages the directory tree and metadata of related files for the HDFS. It could also own a secondary NameNode that can be employed to backup mirror files, combine logs and mirror files periodically and send them back to NameNode. In general, NameNode and Secondary NameNode are deployed on the master node. In addition, the DataNode of HDFS is responsible for storing data and sending processed data back to NameNode and is usually deployed on the slave node.

### 3.2.2. Apache Spark Setup

Apache Spark is installed on top of Hadoop. Spark [21] is a fault-tolerant and distributed data analytics tool capable of implementing large-scale data-intensive applications on commodity hardware. Hadoop and other technologies have already popularized acyclic data flow techniques for building data-intensive applications on commodity clusters. However, these are unsuitable for applications that reuse a working dataset for multiple parallel operations. Some of these applications are iterative machine learning algorithms and interactive data analysis tools. Spark addresses these problems, and is also scalable and fault-tolerant. To accommodate these goals, Spark introduces data storage and processing abstraction called RDDs.

Resilient Distributed Dataset is a collection that has been distributed all over the Spark cluster [22]. RDDs' main purpose is to support higher-level, parallel operations on data in a straightforward manner. Spark can run tasks up to 100 times faster, when it utilizes the in-memory computations and 10 times faster when it uses disk than traditional map-reduce tasks. Spark performs well in these cases, where Hadoop users have reported deficiency with MapReduce. The features like the optimized parameter for iterative jobs using gradient descent and the interactive analytics interfaces used to run queries on large data sets using Hadoop. The behavior of the First Come First Serve (FCFS) scheduling algorithm on a single processor and the gain in speedup with an increasing number of processors used for our experiment is designed as seen in Table 3.

**Table 3.** Experimental Setup.

| Environment | Setup |
|---|---|
| Jobs | Arrival time and CPU burst time |
| Granularity level | Process |
| Type | Fully parallel |
| Expected Speedup | Linear |
| Scheduling Algorithm | FCFS |
| Computing Nodes | Exclusive |
| Process level parameters | Individual process, virtual memory |
| Overall performance parameters | Speedup in latency |
| No. of queues | One |
| Language Support | Scala, Java, Python, R, & MapReduce |

## 4. Experimental Evaluation and Results

In this section, we present the experimental design and results to determine a more efficient option. While experimentation, careful selection of the cluster type, installation, and collection of necessary data help to get a better result. Two clusters are selected for the experiment that could represent different HPC applications. Hence, DeepStack and HPDA clusters are selected in this study for compute-bound and memory-bound jobs.

### 4.1. DeepStack Cluster

The benchmark experiments on the HPC cluster have passed through the test and debugging level programs to test the availability of the cluster server compute nodes and then data with intermediate complexity including Markov chains, Monte-Carlo simulation, distributed PageRank algorithm, HITS algorithm, Distributed Gradient descent. After successful tests of these stages benchmarking programs are used. Benchmark programs include the problems based on highly optimized libraries for core scientific research with competitive baseline figures in the form of linear algebra, genetics, and simulation studies using LPACK, LINPACK, BLAS, etc.

Based on the theoretical basis, the experiment is tested for speedups. Three types of speedups were used in the experiment, and comparable latency and throughput can be achieved using our cluster architecture:

1. linear speedup(fully parallel case experimented),
2. sublinear speedup(sequential parallel mixed case),
3. superliner speedup(cached case—future work)

Items (1) and (2) can be explained by Ahmdal's diminishing returns, whereas (3) is explained by the principle of locality of data (i.e., distributed caching architecture is used). To perform the experiments and compare the results based on the theoretical basis, the sample workload taken for experimentation is seen in Table 4. The latency (*L*) of architecture is the time taken per unit of workload and is given by the following formula:

$$L = \frac{W}{T} \tag{1}$$

where, *T* is the total time taken by the workload on this architecture, and *W* denotes the total workload in the number of instructions/jobs. From the relative performance of two systems processing the same problem as speedup, we measured the speedup/performance improvements from speedup latency and speedup throughput perspectives. Hence, the throughput is the execution rate of a task as follows: density $\rho$, the number of processors. In addition, *Q* is inversely proportional to the latency of the architecture:

$$Q = \frac{\rho \cdot AW}{T} = \frac{\rho \cdot A}{L} \tag{2}$$

where *A*—is the number of processors, $\rho$—execution density( the number of stages in an instruction pipeline, *W*—denotes the total workload executed, and *T* represents the total time taken.

Regarding the speedup in latency, by making the architecture parallel, we intend to speed up the system latency. The following formula is used to compute the speed up in latency between architectures 1 and 2:

$$S_{latency} = \frac{L_1}{L_2} = \frac{T_1 W_2}{T_2 W_1} \tag{3}$$

where *S* denotes the speedup in latency from architecture 1 to 2. In addition, $L_1$ is the latency on architecture 1, and $L_2$ is the latency on architecture 2.

The following formula defines the speedup in throughput:

$$S_{throughput} = \frac{Q_2}{Q_1} = \frac{\rho 2 A_2 T_1 W_2}{\rho 1 A_1 T_2 W_1} = \frac{\rho 2 A_2}{\rho 1 A_1} S_{latency} \tag{4}$$

where $S_{throughput}$ represents the speedup in throughput of Architecture 2 with respect to Architecture 1. Moreover, $Q_1$ denotes the throughput of architecture 1, and $Q_2$ is the throughput of architecture 2.

Regarding Ahmdal's law, if some part of the program cannot be parallelized because of dependencies, then the actual speedup cannot be linearly scaled in proportion to the number of newly added processors. In addition, the speedup is inversely proportional to the amount of sequentially in the program with the following formula:

$$S_{latency}(S) = \frac{1}{(1 - p) + (p/s)} \tag{5}$$

where $S_{latency}$ is the theoretical speedup of the execution of the whole task, $s$ is the speedup of part of the task that benefits from improved system resources, $p$ is the proportion of execution time that the part benefiting from improved resources originally occupied.

**Table 4.** Sample Workload.

| SNo. | Job Id | Arrival Time | CPU Burst Time |
|------|--------|--------------|----------------|
| 1 | P1 | 4 | 200 |
| 2 | P2 | 10 | 500 |
| 3 | P3 | 6 | 400 |
| 4 | P4 | 2 | 300 |
| 5 | P5 | 5 | 200 |
| 6 | P6 | 3 | 100 |
| 7 | P7 | 12 | 250 |
| 8 | P8 | 14 | 320 |
| 9 | P9 | 20 | 250 |
| 10 | P10 | 22 | 150 |

The performance of the DeepStack cluster increased as the compute nodes were added. Theoretical improvement in performance should scale linearly for fully parallelizable tasks, but in a real implementation, a 10% loss in speedup on two-node and a 15% loss in speedup occurred for the three-node cluster. This loss was entirely due to the latency of the network (Ethernet) cable and switch performance. Adding more nodes in a cluster increases the speedup in latency and TAT, decreases the average waiting time of individual processes, and increases communication cost.

In linear speedup, per Ahmdal's law on a fixed parallelizability level in a program, if we increase the number of processors, Torque exhibits an approximately linear increase in speedup as we add new processors.

For the sublinear speed up, if the program has an s%(<100%) sequential part, it exhibits sublinear speedup on Torque. The exact penalty can be computed by Ahmdal's equation and implementation of the program. During experimentation, single compute node, FCFS scheduling performance is used as a benchmark which has execution order as given in Table 5 below and the average baseline performance is seen in Table 6. Note: AT: Arrival time, BT: Burst time, CT: completion time, TAT: Turn around time, and WT: waiting time.

**Table 5.** Torque job execution order on a single node.

| P4 | P6 | P1 | P5 | P3 | P2 | P7 | P8 | P9 | P10 |
|----|----|----|----|----|----|----|----|----|-----|

**Table 6.** Torque job execution order on a single node.

| Process | AT | BT | Start | CT | TAT | WT |
|---------|-----|-----|-------|------|--------|--------|
| P4 | 2 | 300 | 2 | 302 | 300 | 0 |
| P6 | 3 | 100 | 302 | 402 | 399 | 299 |
| P1 | 4 | 200 | 402 | 602 | 598 | 398 |
| P5 | 5 | 200 | 602 | 802 | 797 | 597 |
| P3 | 6 | 400 | 802 | 1202 | 1196 | 796 |
| P2 | 10 | 500 | 1202 | 1702 | 1692 | 1192 |
| P7 | 12 | 250 | 1702 | 1952 | 1940 | 1690 |
| P8 | 14 | 320 | 1952 | 2272 | 2258 | 1938 |
| P9 | 20 | 250 | 2272 | 2522 | 2502 | 2252 |
| P10 | 22 | 150 | 2522 | 2672 | 2650 | 2500 |
| | | | **Avg. Baseline** | **1443** | **1433.2** | **1166.2** |

The cluster-level performance of DeepStack with a 2-node cluster and DeepStack with a 3-node cluster and the expected speedup with the actual gain from the setup is also seen in Table 7 and Table 8, respectively.

**Table 7.** Torque job execution order on 2-node cluster.

| | DeepStack | Speedup | Expected Speedup |
|-----------|-----------|---------------------------|------------------|
| Avg. CT | 804.315 | 1443/804.315 = 1.79 | 2 |
| Avg. TAT | 791.818 | 1.8 | 2 |
| Avg. WT | 525.81 | 2.12 | NA |
| Avg. Latency | 8.28 | NA | NA |

**Table 8.** Torque job execution order on 3-node cluster.

| | DeepStack | Speedup |
|-----------|-----------|---------------------|
| Avg. CT | 583.75 | 1443/583.75 = 2.47 |
| Avg. TAT | 573.8 | 2.49 |
| Avg. WT | 301.5 | 3.87 |
| Avg. Latency | 8.25 | NA |

As per Amdahl's law in this case the speedup in CT and TAT should be twice as of the single processor benchmark but due to internal latencies, it is decreased by 10% for the 2-node cluster and by 15% for the 3-node cluster as seen on Table 9. This is because the performance of clusters declines as described by Amdahl's law. This happens because, there is a delay from locking of processes by a scheduling algorithm used, high internal bandwidth consumption, and poor performance of Ethernet cable capacity. The use of Gigabits per second Ethernet platform among Pbs_servers and compute nodes helps to get better performance.

**Table 9.** Performance Comparison of DeepStack Cluster with Single Node cluster.

| Metric | 1 Node | 2 Node | Speedup | 3 Node | Speedup |
|--------------|--------|---------|---------|--------|---------|
| Avg. CT | 1443 | 804.315 | 1.79 | 583.75 | 2.47 |
| Avg. TAT | 1443.2 | 791.818 | 1.8 | 573.8 | 2.49 |
| Avg. WT | 1166.2 | 525.81 | 2.12 | 301.5 | 3.87 |
| Avg. Latency | 0 | 8.28 | NA | 8.25 | NA |

### 4.2. HPDA Cluster

As part of this study, the evaluation was conducted on the HPDA cluster, running a WordCount program that counts the number of words specified in a given large text file and calculates *pi*. The following are benchmark applications on the HPDA cluster:

- iterative Jobs,
- interactive Analytics,
- distributed Machine Learning,
- streaming Analytics.
- distributed Graphs Processing.

Figure 3 depicts the processing time on a single node and multinode clusters.

The execution time depends on the network communication (I/O), the number of search operations of words, size of the input file. The whole process of MapReduce processes and builds a cost function that explicitly models the relationship between the volume of input data, available system resources (map and reduce slots), and complexity of the reduce function for the target MapReduce job.

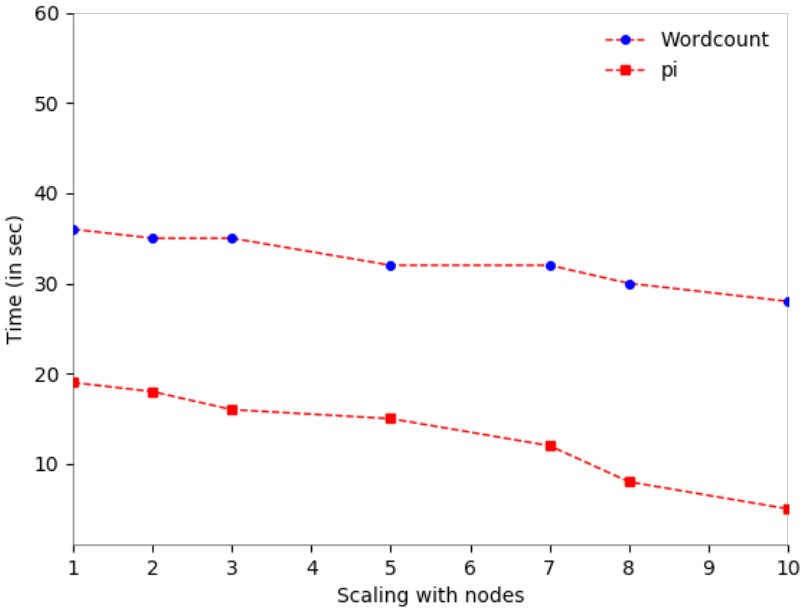

**Figure 3.** HPDA performance Comparison.

The job execution order is compared with a single compute node for base-line comparison. Single Compute Node, FCFS scheduling performance is used as a benchmark for this experiment and the performance result is seen in Table 9.

As per Amdahl's law, in this case, the speedup in compute time (CT) and total time (TAT) should be twice as of the single processor benchmark. But due to Ethernet latencies it is decreased by 15%. Similarly, we have increased the number of compute nodes in the experiment and computed the performance parameters in each setup. Since workload and other conditions are fixed we can use speedup as a measure of performance of our HPC cluster.

From the Hadoop cluster setup of 10 nodes, we were able to aggregate a distributed file system of 6.45 terabytes (TB) from 1 master and 9 live workers running. From this cluster setup, we get 24.9 GB of memory which is enough to run medium size data-intensive applications or jobs. On this cluster, the authors run a WordCount application of 1.23 GB on HDFS and observed when the file is replicated to the worker nodes as this is based on the size of the input file.

Similarly, the researchers were also able to run the finding *pi* application on the HPDA cluster and result that the submitted WordCount application used 72 cores and successfully processed in 5 s only as seen in Table 10. We have run the above-mentioned applications i.e., WordCount and *pi* on a single machine that has 4 cores and 4 GB RAM. Then, the WordCount application successfully processed in 36 s, and *pi* application processed in 19 s which is been shown in Table 10.

**Table 10.** Performance Comparison of HPDA Cluster with Single Node Cluster.

| Cluster Type/App | WordCount App | *pi* App |
|---|---|---|
| Single Node Cluster (4 GB RAM, 1 TB HDD) | 36 s | 19 s |
| Multi-Node Cluster (24.9 GB RAM, 6.4 TB HDD) | 30 s | 5 s |

It has been observed from Table 10, that the execution time depends on the network communication (I/O), the number of search operations of words, size of the input file. The whole process of MapReduce processing and building up a cost function that explicitly models the relationship between the amount of input data, the available system resources (Map and Reduce slots), and the complexity of the Reduce function for the target MapReduce job.

*4.3. Result Analysis*

OpenMP, MPI, and MapReduce are the most widely recognized parallel or distributed programming frameworks. The performance study of three parallel programming frameworks was done [23,24]. The comparative studies have been conducted for two problem sets the all-pairs-shortest-path problem and a joining problem for large data sets. OpenMP [21] is the defacto standard model for shared memory systems, MPI [25] is the defacto standard for distributed memory systems, and MapReduce [24,26] is recognized as the defacto standard framework intended for big data processing. For each problem, the parallel programs have been developed in terms of the three models, and their performance has been observed. The experiment results concluded that if a problem is small enough to be accommodated and the computing resources such as cores and memory are sufficient, OpenMP is a good choice. When the data size is moderate and the problem is computation-intensive, MPI can be considered the framework. When the data size is large and the tasks do not require iterative processing, MapReduce can be an excellent framework. OpenMP is the easiest to use because there is no special attention needed to be paid. After all, it just needs to place some directives in the sequential code. MapReduce is relatively easy to use once we can abstract an application into Map and Reduce steps. The programmers do not have to consider workload partitioning and synchronization. MapReduce programs, however, take a considerable time for the problems requiring many iterations, like the all-pairs shortest-path problem. MPI allows more flexible control structures than MapReduce; hence, MPI is a good choice when a program is needed to be executed in a parallel and distributed manner with complicated coordination among processes.

As per the research questions, we have analyzed the role of commodity hardware in cluster establishment and showed that at minimum an HPC cluster can be configured with an Ethernet switch and cabling inter-connectivity among available commodity computers. We have shown the optimum configuration of the HPC cluster system for various types of workloads and the choice of software and libraries is highly influenced by the type of parallelisms available in application-level programs. For shared-memory programming, we have identified that Torque was a good choice, for distributed memory scatter-gather pattern it was Hadoop based HPC cluster which showed better performance and for distributed in-memory computing it was a spark cluster. Also with the help of the MPI library, Torque supports distributed memory programming paradigm. Lastly, we have analyzed the performance of various clusters under variable load conditions, and we have

designed rather simple experiments which are limited to computing the latency profile of various clusters implemented.

## 5. Related Work

In addition to advances in hardware and communication infrastructure, the HPC community has witnessed a growing set of cluster management solutions, including Torque, Apache Hadoop, Open-Mosiac, Rocks, OSCAR, OpenPBS, Alchemy, and HTcondor [3,10,21,23,27–30]. Several open-source libraries exist, such as OpenCV, CUDA, OpenMP, OpenMPI, MPICH, and many implementations of the message passing (MPI) programming paradigm are on almost all reviewed cluster platforms [21,31]. The libraries for individual research domains, corresponding benchmarks, and quantitative performance evaluations of benchmark problems are also available for most of the HPC implementations [16,32].

In this race of cost-effective HPC implementations, various universities have developed a dedicated HPC clusters. Data centers provide researchers access to HPC resources for cutting-edge research [3,13,17–19] where others have tried to introduce the use of low cost HPC cluster from inexpensive hardware. However, these may not be exhaustively tested for different applications that will affect the performance of the cluster machines [33]. The authors' team believes this represents the importance of infrastructure implementation in academic sectors with relatively fewer costs while getting a nearly equal performing HPC infrastructure for their research activities . The University of Columbia implemented a 167-node cluster with 2672 cores on Dual Intel E5-2650v2 Processors(2.6 GHz) with a Torque/Moab job scheduler in 2009, which was upgraded to support research projects in various application areas [17]. Similarly, Yale University provides an HPC-based computing environment with an excellent publication records. The Stanford HPC center provides a million core compute nodes [18], and the FSU HPC cluster center has more than 10,000 cores and 201,449 Gflops, with a 3 million job capacity [3,10,19,34–36]. Looking to this achievements one can raise a question that, if having HPC can help to solve many research questions, how can I develop HPC by myself?. This question should be answered by someone that knows how. Therefore, we wanted to contribute to answer this question through designing HPC from a cheap computers around us so that others will do the same to solve their own problems.

There is no HPC platform that can be used for varying research activities at the campus level that can be used by the researchers for research that satisfies their needs. This is a bottleneck for researchers in science, engineering, and biological and genetic simulation research [6]. In addition to these domains, core computer science research also requires a specialized HPC centers with commodity computers available in laboratories to research new domains, such as big data, distributed machine learning, the internet of things, and cloud computing [2,3,37].

The parallel or distributed programming frameworks OpenMP, MPI, and MapReduce are the most widely recognized, and the performance of these three parallel programming frameworks with comparative studies has been assessed for two problems sets: the all-pairs-shortest-path problem and the join-problem for large data sets [23]. OpenMP is the defacto standard model for shared memory systems, and MPI [25] is the defacto standard for distributed memory systems. Finally, MapReduce [26] is recognized as the defacto standard framework for big data processing.

For each problem, parallel programs have been developed regarding the three models, and their performance has been observed. The experimental results indicated that if a problem is small enough to accommodate sufficient computing resources, such as cores and memory, OpenMP is a good choice. When the data size is moderate and the problem is computationally intensive, MPI can be considered for the framework. When the data size is large and the tasks do not require iterative processing, MapReduce can be an excellent framework.

OpenMP is the easiest to use because no special attention is needed, as it just requires to place some directives in the sequential code. MapReduce is relatively easy to use once we abstract an application into the map and reduce steps, and programmers do not have to consider workload partitioning and synchronization. MapReduce programs, however, take a considerable time for problems requiring numerous iterations, such as the all-pairs shortest-path problems. Moreover, MPI allows more flexible control structures than MapReduce; hence, MPI is a good choice when a program must be executed in a parallel and distributed manner with complicated coordination among processes.

The language-independent MPI is a communications protocol for parallel computing where point-to-point and collective communication are supported [26]. However, the standard does not currently support fault tolerance [25] because it primarily addresses HPC problems. Another MPI drawback is that it is unsuitable for the small-grain level of parallelism, for example, to exploit the parallelism of multicore platforms for shared memory multiprocessing. In contrast, OpenMP is an Application Programming Interface (API) that supports multi-platform shared memory multiprocessing programming on most processor architectures and operating systems [21]. OpenMP is becoming the standard for shared memory parallel programming due to its high performance; however, it is unsuitable for distributed memory systems. The idea of extending this API to cope with this issue is now a growing field of research [38]. OpenMP's user-friendly interface allows it to easily parallelize complex algorithms, unlike MPI, because the code must be heavily re-engineered to obtain relevant performance improvements.

Spark is a state-of-the-art framework for HPC designed to efficiently deal with iterative computational procedures that recursively perform operations on the same data [39,40], such as supervised machine learning algorithms. Spark is based on the concept of maintaining data in memory rather than on disk, as done by other well known approaches such as Apache Mahout, which require data reloading and incur considerable latencies. Experiments have shown that Spark outperforms conventional MapReduce jobs in terms of speed by up to two orders of magnitude [20,40,41]. The core data units in Spark are called resilient distributed datasets (RDDs). They are a distributed, immutable, and fault-tolerant memory abstraction that collects an element set in which an operation set can be applied to produce other RDDs (transformations) or return values (actions). The RDDs can reside in memory, on the disk, or a combination of these. However, they are only computed on actions following a lazy evaluation strategy to perform minimal computation and prevent unnecessary memory usage. The RDDs are not cached in memory by default, therefore, when data are reused, a persist method is needed to avoid re-computation.

Various cluster management options are available for running Spark. The options range from the simple Spark integrated Stand-alone Scheduler to other widespread cluster managers, such as Apache Mesos and Hadoop YARN [30,42,43]. To get benefited from this reach features, this study deploys Spark in a Hadoop cluster. Apache Hadoop is an open-source software platform for distributed big data processing over commodity cluster architectures [30,44]. It has three main elements: (a) a MapReduce programming model that separates data processing into mapping to perform data operations locally, shuffling to redistribute network data and reduce data summarization; (b) a Hadoop distributed file system (HDFS) with high-throughput data access; and (c) a cluster manager (YARN) handling available computing resources and job scheduling. Nevertheless, Spark on Hadoop may be preferred [30,43] because it also

- offers a distributed file system with failure and data replication management.
- allows the addition of new nodes at run time, and
- provides a set of tools for data analysis and management that is easy to use, deploy and maintain.

The authors identified specific gaps in the HPC platform that can easily developed from commodity computers around us for research activities. To our knowledge, there is no HPC developed from cheap computers to serve a fully functional HPC infrastructure the can be used for any research activities ranging from simple simulation to a very

complicated computation that requires high computing power.There are attempts to design HPC from low cost machines and Raspberry Pi [33,45] but the first approach didn't test the environment for range of job types where the second mainly compared the energy consumption of their approach with the commodity servers. Authors from [46] also applies similar concept yet they designed the cluster for odd-even sorting which cannot represent all ranges of applications. Therefore, to address this technical gaps and come up with comprehensive solution, we designed a cost-effective HPC cluster from commodity computers around us. The novelty of this approach is, we put together the best existing programming language independent tools that any one can easily access and state-of-the-art frameworks to deal with it. Hence we implemented, tested, and compared our scheme that helps interested researcher to develop and use HPC from commodity computers.

## 6. Conclusions

The study was conducted with the available commodity hardware and open-source tools. The study team developed two types of clusters. Depending on the type and complexity of the computation, individual clusters can be chosen to perform jobs. The DeepStack cluster supports second-generation or low-level programming languages, such as C, C++, and Fortran. The HPDA cluster supports the latest programming or high-level programming languages such as Java, Scala, Python, and R.

Many pieces of research indicate that the DeepStack cluster is a good choice if a problem is small enough to be accommodated, the computing resources, such as cores and memory, are sufficient, and the data size is moderate. When the data size is large and the task requires high-speed iterative processing, then the HPDA cluster is a good choice.

From the results, we have analyzed the role of commodity hardware in cluster establishment. We have identified that at minimum an HPC cluster can be configured with an Ethernet switch and cabling interconnectivity among available desktop computers with the help of suitable cluster management software. But, during the experiment, we found that the rate of failure of commodity computers is more as compared to the specially designed computers, also there is less than 50 Mbps bit rate due to various types of delays between compute-node and server-nodes, which is quite less than the rate at which connected hard disk can supply data to a processor using SCSI or PCI.

We also found the optimum configuration of the HPC cluster system for various types of workloads. The choice of software and libraries is highly influenced by the type of parallelisms available in application-level programs. For shared-memory programming, we have identified that Torque was a good choice, for distributed memory scatter-gather pattern it was a Hadoop-based HPC cluster which showed better performance and for distributed in-memory computing, it was a spark cluster. Also with the help of the MPI library Torque supports distributed memory programming paradigm.

Lastly, we have analyzed the performance of various clusters under variable load conditions, but as per the scope of this research, we have designed rather simple experiments which are limited to computing the latency profile of various clusters implemented. More sophisticated experiments involving the performance gain at compiler level optimizations, use of libraries, programming paradigms, and different types of workloads shall be studied in the future.

**Author Contributions:** Conceptualization, D.R. and Y.S.; methodology, D.R. and H.Y.; investigation, Y.S. and H.Y.; validation, H.Y.; visualization, D.R. and Y.S.; writing—original draft preparation, D.R.; writing—review and editing, D.R., H.Y. and Y.S. All authors have read and agreed to the published version of the manuscript.

**Funding:** This work was supported by the National Research Foundation of Korea (NRF) grant funded by the Korean government (MSIT) (No.2021R1C1C1010861). This work was supported in part by the Korea Institute for Advancement of Technology (KIAT) grant funded by Korea government (MOTIE) (P0012724, The Competency Development Program for Industry Specialist) (Corresponding Author: Yongseok Son).

**Institutional Review Board Statement:** Not applicable.

**Informed Consent Statement:** Not applicable.

**Data Availability Statement:** For more, we put the link to the installation manual and configuration steps at the following https://github.com/Derejereg/HPC/blob/main/Installation_manual.pdf (accessed on 27 April 2022) using GitHub.

**Conflicts of Interest:** The authors declare no conflict of interest.

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
