# Peer review of "Harvesting the Aggregate Computing Power of Commodity Computers for Supercomputing Applications"

_applsci, doi:10.3390/app12105113_

Round 1

Reviewer 1 Report

In this second version the quality of the paper has improved.

The authors have attending some of my comments and suggestions. However, there are some open issues, which we list below:

  • In the Introduction the authors refer: “Training and workshops may need to be organized to introduce the available resources to the staff and students at the university to provide basic libraries to support their experiments. Moreover, an easy-to-use manual for loading data sets is expected by novice users.” I agree and is why one of my previous comments was about the text “The installation manual is supplemented with the necessary references and links for various libraries that are needed to be installed on the cluster.”. However, the authors deleted this text and mention “We assumed that this is not a critical part of the paper and ignored it from this article. As you suggested, we can provide details like a link for anyone who needs details for reference based on request.” In my opinion, this is not an adequate attitude, and the authors must give the link to the installation manual. This is critical, in my opinion, because it is one of the main contributions of this work.
  • Also, in the Introduction section, it is mentioned as one of the contributions of this work “We survey of existing cluster solutions at the primitive university level, advanced research laboratory level, and industry-level, such as Google.” In my review I had the following comment: “Where? Moreover, Google is never mentioned in the paper (only in this sentence). And the authors had answered: “We evaluated the existing cluster solutions and compared them with our approach. This helped us to conclude our clusters into either CPU bound or storage bound operations and we tested our designs based on these two environments. We have an intention of mentioning that Google is just an example. But, to accommodate your comment, we removed it since we did not address the example directly in this paper.” That is not correct because Google continues mentioned.
  • Lines 86-95: “The rest of this paper is organized as follows. Section 2 deals with the available experiences through reviews, and Section 3 discusses the main challenges identified in the design, implementation, and post-implementation of the cluster of commodity computers. Section 4 describes the proposed transition model for a single campus with the architecture of commodity/HPC clusters with the expected number of nodes and cores. Moreover, this section addresses the issues of scalability. In this regard, many architecture and implementation models are necessary for software and libraries for major application areas of the cluster to fulfill the gaps identified in this comparative study. Section 5 concludes the paper with recommendations for the use of the output on a similar scale.”. There are many problems with this sentence. First section 2 is not well described according to its contents (Background and Motivation). Second, Section 5 “Related Work” is not mentioned. Third, Section 5 doesn’t conclude the paper but Section 6.
  • Lines 117-118: “Since the development of the supercomputer by their national defense university, the na first win of” ?? Please rewrite the sentence.
  • Lines 143-151: This is almost repeated with the Introduction.
  • The authors mention that questionnaires were prepared to collect experiential and heuristic knowledge of different stakeholders and administrators of existing HPC facilities. However, there is no information about the questions in the questionnaire neither the conclusions achieved.
  • Line 187: “Data Collectin - ”, please correct the typo.
  • In Line 204 it is stated “The actual implementation of three separate types of HPC clusters was initiated within separate initiatives of data science and intelligent systems research groups” Which ones are the three types of HPC clusters? In the Conclusion it is written “The study team developed two types of clusters.”
  • Line 201: “Verification and validaion -”, please correct the typo.
  • Line 214: “Design of experimnts -”, please correct the typo.
  • Line 225: “Data Analyss -”, please correct the typo.
  • Line 230: “Tto save” , please correct the typo.
  • Lines 274-275: “wall-time (elapsed time between the start of the process and ’now’)”. Now? Please rewrite the sentence.
  • FCFS acronym is not defined in the text.
  • Lines 346-347: “The design and use of this platform require aboard of all libraries and implemented on available hardware”. Please rewrite this sentence.
  • Lines 405-406: “This is because, on top of the nature of the performance decline of clustered processors …” Please rewrite this sentence.
  • Lines 436-437: “The cluster able to compile the memory of 24.9GB,” Please rewrite this sentence.
  • Line 462: “The experiment results concluded that If a”, please correct the typo.
  • Line 522: “for big data processing. r each problem”, please correct the typo.
  • … (Many typos)

I continue recommending that a native-English-speaker review this paper.

In conclusion, the paper can be improved provided that the authors answer the above-mentioned questions and modify the paper according to the suggestions.

Reviewer 2 Report

This paper has been revised according to the reviewers' comments and presented in a good manner.  Suggest to accept it after English spelling check.

Author Response

This manuscript is a resubmission of an earlier submission. The following is a list of the peer review reports and author responses from that submission.

Round 1

Reviewer 1 Report

GENERAL OVERVIEW

In this paper, the authors design and implement an HPC-based supercomputing facility from commodity computers at an organizational level to provide two separate implementations for cluster-based supercomputing using Hadoop and Spark-based HPC clusters, primarily for data-intensive jobs and Torque-based clusters for Multiple Instruction Multiple Data(MIMD) workloads.

The performance of these clusters is measured through some experiments.

WEAKNESSES

- Which is the innovation of this paper?

- Which differentiates this work from the others mentioned in Related Work section?

- Which are the main contributions of this work? The authors mention at the Introduction section:

“We analyze the existing HPC infrastructure design options and identify significant challenges related to the hardware, software, and network in the implementation of the cluster computing.” Where?

“we conduct a survey of existing cluster solutions at the primitive university level, advanced research laboratory level, and industry-level, such as Google.” Where? Moreover, Google is never mentioned in the paper (only in this sentence).

“introduce commodity cluster to aggregate the computing power of laboratory computers and use an efficient scheduler to run jobs on the cluster infrastructure.”

“We present a model for the step-by-step design and implementation of cluster of commodity computers in a laboratory environment.” Where is the model?

The authors should clarify these contributions in the text.

- Section 4. Experimental Evaluation and Results should be reformulated. The authors must clearly define the experimental setup. Which are the benchmarks used in each experiment and why? Which is the size of datasets? It would be interesting to also use some metrics such as CPU and memory used.

The authors mention: “After succesfull test of these stages benchmarking programs are used.”  Which benchmarks?

After that is refereed “Benchmark programs include the problems based on highly optimized libraries for core scientific research with competitive baseline figures in the form of linear algebra, genetics, and simulation studies etc. using LPACK, LINPACK, BLAS etc. Implementation of these libraries is proposed to be in second round of this project.” Please explain what is used in the experiments.

- “The throughput is defined on the execution density ρ, the number of processors.” Please clearly define throughput.

- Table 4 Sample Workload is not explained neither referenced in the text.

- Table 5. Single and Performance Comparison appears in page 8 before Table 4. However, it is only referenced in page 11.

- Section 2 Background and Motivation should be expanded and improved. The motivation for this work should also be well defined.

The methodology used in this work is not well explained. The authors only refer: “We followed the standard methodology for information technology/information and communication technology projects during the implementation of DeepStack and high-performance-data analytics (HPDA) clusters.” Which standard methodology? Please give also a reference.

After that is mentioned: “This study used a hybrid methodology involving qualitative, quantitative, and empirical research aspects at various stages.” Give more details about this “hybrid methodology”.

- In page 4 is mentioned: “The list of all dependencies and libraries in our implementation is given in the appendix,…” However, the paper has no appendix.

The authors also refer: “The installation manual is supplemented with the necessary references and links for various libraries that are needed to be installed on the cluster”. In this case it will be useful to have a link to the installation manual.

- In page 4 we have: “The DeepStack cluster architecture is given below.” The authors must reference as Figure 1.

- In the text the authors reference the authentication server module as “TRQATHD”, which is not according to Figure 1.

- “Wall-time” should be defined in page 5.

- It is mentioned that “Spark can run tasks up to 100 times faster, when it utilizes the in-memory computations and 10 times faster when it uses disk than traditional map-reduce tasks.” Therefore, this should be proved with the experiments.

- Related work section must be upgraded with more recent works. The most recent work in the references is from 2016 !

The authors must also compare this work with the related works.

- Which are the main scientific conclusions of this work? Please pointed it out in the last section.

As a general comment it is kindly recommended that native-English-speaker for linguistic improvements review this paper.

MINOR COMMENTS

- In Section “4. Experimental Evaluation and Results” the authors have the following subsection “4.1. DeepStack Cluster” without text between them. I suggest giving a small introduction to the contents of section 4 before start subsection 4.1.

- Lines 209-210: “Each set of sub-clusters contains one server, an instance of scheduler, one authentication demon at server.” should be “Each set of sub-clusters contains one server, an instance of scheduler, one authentication daemon at server.”

- Line 283: “Regading Ahmdal’s law,...” should be “Regarding Ahmdal’s law,...”.

- Line 294: “This loss was intirely …” should be “This loss was entirely …”

- Lines 347-349: “Then, the WordCount application successfully processed in 36 seconds and pi application processed in 19 seconds which is been shown in Table” should be “Then, the WordCount application successfully processed in 36 seconds and pi application processed in 19 seconds which is been shown in Table 10.”

- All the lines of “Table 1 The specific modelling tools in the simulation experiment.” should be in the same page.

In conclusion, the paper can be improved provided that the authors answer the above-mentioned questions and modify the paper according to the suggestions

Reviewer 2 Report

In this paper, it discussed how to setup an system platform to supercomputing applications with two types of clustering based modes. It clearly explained how to set up the system with hardware and software as well as the device standards. It seems as a technical report rather than a technical paper. I did not see what the key innovation points are. What is the main advantages over the other supercomputing platforms doing the similar tasks. Another point is that the main contributions listed by authors may not include something new. In summary, I can not recommend to accept it.

Reviewer 3 Report

Thanks for this paper, this research created 2 clusters: DeepStack and HPDA with commodity computers, but this draft is not enough for consideration of a  journal publication. Most stuff in this draft is the introduction of some open-source tools, e.g. Torque and Hadoop, and how to use them. The testing part only includes some preliminary results, e.g. speedup, performance loss due to inter-node comm, but didn't explain the reasons in detail. Also, the third architecture (superlinear speed (cache case)) is really necessary for a complete testing. I think it would be better to fill in more research content before resubmitting.